# Neuroimaging Techniques as Potential Tools for Assessment of Angiogenesis and Neuroplasticity Processes after Stroke and Their Clinical Implications for Rehabilitation and Stroke Recovery Prognosis

**DOI:** 10.3390/jcm11092473

**Published:** 2022-04-28

**Authors:** Lidia Włodarczyk, Natalia Cichon, Joanna Saluk-Bijak, Michal Bijak, Agata Majos, Elzbieta Miller

**Affiliations:** 1Department of Neurological Rehabilitation, Medical University of Lodz, Poland Milionowa 14, 93-113 Lodz, Poland; 2Biohazard Prevention Centre, Faculty of Biology and Environmental Protection, University of Lodz, Pomorska, 141/143, 90-236 Lodz, Poland; natalia.cichon@biol.uni.lodz.pl (N.C.); michal.bijak@biol.uni.lodz.pl (M.B.); 3Department of General Biochemistry, Faculty of Biology and Environmental Protection, University of Lodz, Pomorska, 141/143, 90-236 Lodz, Poland; joanna.saluk@biol.uni.lodz.pl; 4Department of Radiological and Isotopic Diagnosis and Therapy, Medical University of Lodz, 92-213 Lodz, Poland; agata.majos@umed.lodz.pl

**Keywords:** stroke, recovery, neurorehabilitation, neuroimaging, neuroplasticity, angiogenesis

## Abstract

Stroke as the most frequent cause of disability is a challenge for the healthcare system as well as an important socio-economic issue. Therefore, there are currently a lot of studies dedicated to stroke recovery. Stroke recovery processes include angiogenesis and neuroplasticity and advances in neuroimaging techniques may provide indirect description of this action and become quantifiable indicators of these processes as well as responses to the therapeutical interventions. This means that neuroimaging and neurophysiological methods can be used as biomarkers—to make a prognosis of the course of stroke recovery and define patients with great potential of improvement after treatment. This approach is most likely to lead to novel rehabilitation strategies based on categorizing individuals for personalized treatment. In this review article, we introduce neuroimaging techniques dedicated to stroke recovery analysis with reference to angiogenesis and neuroplasticity processes. The most beneficial for personalized rehabilitation are multimodal panels of stroke recovery biomarkers, including neuroimaging and neurophysiological, genetic-molecular and clinical scales.

## 1. Introduction

Stroke is an acute cerebral, spinal, or retinal vascular accident with neurological dysfunction that persists longer than 24 h, or one of any duration when infarction or hemorrhage corresponding to symptoms is demonstrated by imaging (computed tomography/ magnetic resonance scans) or autopsy [1]. The clinical symptoms are very heterogenous and conditional on the topography of damage [2]. There is effective, specific treatment for acute ischemic stroke available, such as thrombolysis (intravenous administration of tissue plasminogen activator) or endovascular thrombectomy [3]. Furthermore, ischemic stroke patients should be administrated with antiplatelet or anticoagulant drugs (depending on the etiology of ischemia) as secondary stroke prevention [4]. The procedure for acute hemorrhage stroke is based on intensive blood pressure and intracranial pressure reduction. In addition, early open-surgery evacuation or drainage of hematoma might be beneficial and should be considered in specific cases [5]. Following the onset of stroke, natural, spontaneous processes of recovery occur, but are generally incomplete and difficult to predict amongst individuals. By fostering angiogenic and neuroplastic changes, restitution of function in damaged, nearby, or distant (but functionally or structurally connected) regions to the lesion may become more efficient [6]. Therefore, novel therapeutic approaches of stroke rehabilitation should be a focus of attention among researchers. Neurorehabilitation is intended to aid stroke patients to become as independent as possible and should be introduced soon after the patient’s condition stabilizes. Novel rehabilitation strategies are specific to the individual’s therapeutic goal, so directed at affected functional areas including motor impairments (most frequent), gait disturbance, speech disorders, cognitive failure, vision disturbances, etc. Therefore, it is appropriate to provide post-stroke rehabilitation by an interdisciplinary medical team together with physicians, psychiatrists or psychologists, neurologists, physical and occupational therapists, speech-language pathologists, nutritionists, and others [7]. Increasingly, novel technology applications are employed in post-stroke procedures, such as rehabilitation robots—as therapy devices (to train lost motor function) as well as assistive devices (to compensate lost skills) [8]. When considering the strategies of modern post-stroke rehabilitation, the importance of non-invasive neuromodulation techniques including repetitive transcranial magnetic stimulation and transcranial direct current stimulation should also be emphasized [9,10,11,12]. Furthermore, virtual reality programs on mobile devices may constitute a time-efficient, clinically effective, easy-to-implement and goal-oriented tool for upper extremity stroke rehabilitation [13]. A relevant issue in the context of this article is that the implementation of neuroimaging measurements in clinical studies will enable a multimodal approach to brain investigation and prediction of functional recovery after stroke. Indeed, neuroimaging and brain mapping provide a wide perspective on brain repair by showing modulations in cortical and subcortical activity or changes in functional brain connectivity [14]. The aim of this article is to review the literature and analyze neuroimaging measurements and their role in predicting recovery after stroke. Simple brain imaging measures have proved value in total infarct volume, correlating with predicting overall neurological status. Saver et al. indicate that subacute CT infarct volume correlates moderately with the 3-month clinical outcome, as determined by widely used neurological and functional assessment scales [15]. Nowadays, there are more complex imaging measures used as biomarkers of recovery. We use them to analyze both structure (anatomical assessment of key pathways injury, particularly corticospinal fiber number by diffusion tensor imaging) as well as function (for example, imaging of region pertinent to specific neurotransmitters or evaluation of cortical excitation) of brain tissue [16]. In this review, we have decided to briefly introduce the field of neuroimaging possibilities and analyze selected neuroimaging biomarkers in the context of recovery processes such as angiogenesis and neuroplasticity.

For outcome prognosis, recovery processes such as angiogenesis and neuroplasticity should be analyzed. Angiogenesis is a renewed growth of blood vessels to restore blood supply to the damaged brain tissue and it occurs after stroke [17,18]. This restorative action is undoubtedly beneficial after stroke; it was proved that there is a stroke-related period of heightened vascular plasticity that is correlated with restoration of blood flow and then predictive of motor function recovery [19]. However, it is also worth remembering that angiogenic factors, such as VEGF and MMPs, contribute to blood brain barrier injury and may lead to oedema formation or haemorrhagic transformation [20]. Neuroplasticity definition contains adaptive structural and functional processes, including synaptogenesis, neurogenesis and neuroprotection. The critical period for post-stroke recovery and neuroplasticity is considered the acute (0–7 days) and the early subacute (7 days-3 months) phase [8]. In the acute phase, secondary neuronal networks are exploited to preserve function, whereas in the subacute stage, new synaptic connections are formed, and in the chronic phase, remodeling by axonal sprouting and then reorganization occurs [21]. There is also enhanced expression of growth associated genes and proteins observed [22], changes in GABA (gamma-aminobutyric acid), NMDA (N-methyl-D-aspartate) receptor subtypes and upregulation of NMDA receptors to increase brain excitability [23]. Worth noticing for this review is also the theory of diaschisis, the assumed function decrease in spatially discrete brain areas, functionally related to the site of injury [24]. Diaschisis can be demonstrated by neuroimaging techniques that evaluate changes in cerebral blood flow, neurotransmitters, and metabolism action in regions distant from the lesion.

## 2. Neuroimaging Techniques Dedicated to Stroke

Neuroimaging is usually associated with differentiating ischemic from hemorrhage stroke in the acute phase; it also has a crucial role in ischemic stroke patients’ selection for novel treatment options, including late window thrombectomy. In this review, we emphasized the role and potential of neuroimaging techniques for post-stroke recovery prediction. Correlating recovery prediction with measurements entitles researchers to use determination of biomarkers. The FDA-NIH biomarker working group presented the following definition of biomarker: “A defined characteristic that is measured as an indicator of normal biological processes, pathogenic processes or responses to an exposure or intervention” [25], considering the harmonization of terms used in science as a requisite priority. As stroke studies develop, the role of term biomarkers has changed from diagnosis to therapeutic mechanisms and currently includes a wide group of factors with various origin: genetic, molecular, clinical scales, neuroimaging and neurophysiological. Figure 1 shows the emerging roles of stroke biomarkers [26]. Figure 2 presents a classification of neuroimaging techniques with potential for use in stroke recovery prognosis. Neuroimaging stroke recovery biomarkers include measures of structure and function [16]. To evaluate structure, we commonly use characteristics including infarct volume, extent of cortical or white matter injury, white matter integrity, and percentage of corticospinal tract injury. Functional assessment should be focused on features including activation within ipsilesional and contralesional sites, interhemispheric balance, resting state functional connectivity, task-based synchronization, and desynchronization, as well as cortical excitability, facilitation, and inhibition [27]. In the next section, we briefly discuss techniques dedicated to stroke recovery analysis with reference to angiogenesis and neuroplasticity processes. Table 1 summarizes the techniques discussed in the text.

## 3. Stroke Recovery Prognosis Based on Selected Neuroimaging Measurements—Review of Literature

### 3.1. Potential Angiogenic Biomarkers of Stroke Recovery

Angiogenesis after stroke is a complex and multistep process, involving (after gene transcription and releasing proangiogenic factors) endothelial cell proliferation, vascular sprouting, and finally microvessel formation [28]. In the current state of knowledge, imaging techniques allow evaluation of a wide spectrum of structural and functional tissue features. Recent developments in magnetic resonance imaging (MRI) techniques presented the possibility of assessing tissue perfusion and estimating the quantification of various features in the vascular network, including microvascular cerebral blood volume (CBV) or microvascular density [29]. In an experimental study of Yanev et al., steady-state contrast-enhanced (ssCE-) MRI with a long-circulating blood pool was applied to characterize the model of vascular reorganization within the ischemic lesion and in secondarily affected areas, from subacute to chronic stages, after focal cerebral stroke [30]. Results showed that dynamic revascularization in the perilesional area and progressive neovascularization in non-ischemic connected areas were found. These phenomena may contribute to non-neuronal tissue remodeling and be relevant in post-stroke recovery. In turn, the initial stage of angiogenesis might be observed by MRI techniques as blood-brain barrier leakage, because BBB permeability is associated with endothelial cell proliferation and vascular sprouting [31]. To quantify BBB integrity, the dynamic contrast-enhanced MRI (DCE-MRI) with gadolinium chelates could be included in patients with changes in MRI signals, which occur as a consequence of leakage of an intravenously injected contrast into the interstitial area [32]. According to Pradillo et al., DCE-MRI was shown to be a useful and non-invasive technique for evaluation of vascular function and angiogenesis processes [33]. The most prevalent investigation into direct assessment of vessels is angiography MR. However, the direct depiction of cerebrovascular remodeling or angiogenesis is very rare in clinical usage, because of the relatively low spatial resolution of MRI. Endothelial sprouting and microvessel formation proceed at length scales that are at minimum one order of magnitude lower than the current technology provides. Nevertheless, in the recent experimental study of Kang et al., high-resolution T1-contrast based on ultra-short echo time MR angiography (UTE-MRA) was carried out to visualize macro- and microvasculature and their association with ischemic edema status in transient middle cerebral artery occlusion rat models [34]. Although the presented results are very promising, the limitations associated with MRI as a tool for demonstrating post-stroke angiogenesis should be taken into consideration. Elevation of CBV is not only specific for neovascularization but may also contribute to the collateral flow and vasodilatation of existing vessels. Likewise, contrast agent leakage may arise in relation to angiogenesis as well as a result of BBB disruption in response to vascular pathology and hemorrhage [31]. To ensure new insights into the role of vascular remodeling in functional recovery after stroke, more studies need to be performed. The use of modern technologies may be very valuable as a monitoring tool for possible future therapies designed to support neovascularization in post-stroke patients, thus providing the possibility of introducing personalized therapies.

### 3.2. Neuroplastic/Neurogenic Markers of Stroke Recovery

In many current studies, neuroimaging techniques are employed to assess and understand post-stroke impairment. Understanding neural mechanisms of brain tissue damage as well as regeneration processes could be essential for predicting recovery and monitoring therapy. Such neural mechanisms of recovery involve in particular the perilesional tissue in the injured hemisphere, but also the contralateral hemisphere, subcortical and spinal regions [35]. All those processes that support recovery, termed neuroplasticity, are possible to identify as structural and functional brain changes in various neuroimaging techniques, such as magnetic resonance imaging (MRI), functional MRI, and functional magnetic spectroscopy (MRS) [36]. What is more, by using neurophysiological agents such as electroencephalography (EEG), we can map the brain activity and by using transcranial magnetic stimulation (TMS), it is also possible to test the influence of specific brain regions on motor learning and post-stroke recovery [37]. Described below are actionable techniques to predict post-stroke recover, that might be used for a personalized strategy of stroke treatment.

MRI is an essential clinical tool for diagnosing stroke severity, implementing treatment and predicting outcome [38]. Multimodal MRI reveals various parameters that help determine stroke mechanisms which affect recovery, such as differentiation of ischemic core from ischemic penumbra. The ischemic core is an area of infarction, which develops rapidly after artery occlusion. The differentiating parameter is the cBV, which is kept in the penumbra zone and decreased in the ischemic core. Using MRI technology, ischemic lesions can be identified with high precision, using diffusion-weighted image (DWI). Perfusion-weighted image (PWI) in turn can identify ischemic penumbral tissue [39]. To assign areas with PWI-DWI mismatch (the area difference when the perfusion lesion is larger than the diffusion lesion) is to identify representative salvageable tissue that may be responsible in recovery [40]. There is agreement as to the usefulness of characterizing the ischemic penumbra at the acute stage in relation to predicting motor outcomes. However, there are also data which suggest that the location of ischemic penumbra, instead of volume, could predict outcome and affect motor recovery [41].

MRI also delivers a method for assessing indices of white matter integrity and remodeling following ischemic stroke through diffusion-based methods. Measures of corticospinal tract (CST) white matter integrity is possible by diffusion tensor imaging (DTI) that uses anisotropic diffusion to estimate the axonal (white matter) organization of the brain. Ratio and asymmetry index of fractional anisotropy (FA) between ipsi- and contralesional corticospinal tracts (CSTs) is a very popular predictor in DTI studies. In general, a lower FA value of the ipsilesional CST may indicate greater damage tp the CST that can lead to more Wallerian degeneration of CST axons [42]. It has been proved that corticospinal tract injury is a valuable predictor of motor recovery in acute and post-acute stages [43,44,45]. In turn, Doughty et al. discovered that FA reduction of the CST (detected in the acute phase of stroke) present fractional predictive value to motor outcomes at 3 months [46]. Essentially, FA value can also be influenced by other factors (not only damage of CST), such as white matter architecture, so it should be carefully considered as a biomarker of brain impairment and poor recovery. Furthermore, according to Cassidy et al., CST-related atrophy and CST integrity are not useful in predicting treatment-related behavioral gains [47]. Only percentage of CST injury significantly predicted motor gains in response to therapy in the setting of subacute-chronic stroke, and might be used to stratify variables in restorative stroke trials. Likewise, Lim et al. proved that CTS injury provides low levels of prediction of upper extremity motor recovery and only in patients with severe initial motor deficits [48]. In a recently published study from Mattos [49], white matter plasticity effected by intensive task-specific upper limb training in chronic stroke was investigated and there were no changes in white matter integrity in patients with motor improvements. There was significant diversity in the response to test-specific training allowing for the specification of responder and non-responder groups. In responders, larger motor recovery was correlated with integrity in contralesional fibers and non-responders had severe damage of transcallosal fibers (more than responders). These results suggest the engagement of interhemispheric processes in responder groups. In many different stroke studies, it has also been proved that brain tissue reorganization in motor tracts and subsystems extends beyond the corticospinal tract [50,51,52].

After stroke damage, we can observe a dynamic process of changing brain activation patterns. Measurement of brain function presents complexities that do not increase with the measurement of anatomy. The way to follow this complexity is functional MRI (fMRI), which measures brain activity by detecting changes associated with blood flow. It is considered that neuronal activation and cerebral blood flows are coupled [53]. The most common form of fMRI is based on the blood-oxygen-level dependent (BOLD) contrast, which can indirectly measure neural activity based on changes in blood flow and deoxyhemoglobin concentration [54]. To activate the brain with fMRI, a specific behavioral paradigm must be executed by a patient; it should be performed on command, correctly and on time. Therefore, the behavioral paradigm should also be carefully selected to investigate the brain’s functional field of interest. However, post-stroke motor impairments can make even simple motor performance difficult; thus, resting-state imaging is an attractive method for studying stroke network activity. With this technique, the functional connectivity represents the synchrony of intrinsic blood oxygen level-dependent (BOLD) signal fluctuations among different brain regions [55]. Functional connectivity that reflects the integrity of various motor and non-motor networks is associated with stroke outcome [56,57]. In a study by Puig, it has also been proved that patients with good outcome had greater functional connectivity than patients with bad outcome [58]. It has been shown by preserved bilateral interhemispheric connectivity between the anterior inferior temporal gyrus and superior frontal gyrus and decreased connectivity between the caudate and anterior inferior temporal gyrus in the left hemisphere. A larger potential for post-stroke recovery is also linked to the dynamic connectivity (time-varying functional network connectivity) between the bilateral intraparietal lobule and left angular gyrus [59]. Dynamic connectivity analysis is relevant in showing transiently increased information exchange between motor domains in moderate motor stroke and more isolated information processing in severe motor stroke [60]. To generalize, it is suggested that a better recovery is linked to keeping or recreating normal brain activation patterns and a bad recovery is linked to persistent contralesional fMRI response [61].

The next technique that allows assessment of the function of brain tissue is MRS (magnetic resonance spectroscopy). The presence and concentration of various metabolites is analyzed based on the principle that the distribution of electrons within an atom cause nuclei in different molecules to experience a slightly different magnetic field. In a study by Blicher et al., the higher GABA level in ipsilesional M1 was related to better motor function improvements after constraint-induced therapy [62]. In a recent report [63], it has been demonstrated that progressive falls in NAA and late increases in choline, myoinositol and lactate may indicate progressive non-ischemic neuronal damage, which has a harmful effect on motor recovery. Undoubtedly more studies linking MRS to functional outcomes would be beneficial in the context of personalized treatment choices.

Electroencephalography (EEG) is the most common, non-invasive method to record spontaneous or evoked electrical oscillation at various frequencies of the brain and, importantly, is one of the few mobile techniques available, unlike CT and MRI. Assessment of neuronal oscillations with electro or magneto-encephalography may supply an easy, available method to evaluate the balance between excitatory and inhibitory cortical actions [64]. EEG signals can identify sensitive changes in brain activity that cannot be detected by clinical measures. Furthermore, quantification of the EEG signal before and after treatment (rehabilitation) may evaluate neuroplasticity near the lesion and within whole-brain networks. The general findings, suggesting bad recovery in post-stroke patients investigated with EEG or MEG at the acute or subacute stages, indicate predominant inhibitory processes in the perilesional areas of cortex, shown by increased low-frequency oscillations [65,66]. Simultaneously, good recovery after stroke is associated with increased sensorimotor excitability in acute stages, presented by lower beta-rebound as a reaction to tactile finger stimulation [67]. Regarding the chronic post-stroke stages, in Mane’s study, changes in the cortical activity after different types of motor rehabilitation, and its possible outcome, were analyzed using quantitative electroencephalography (QEEG) [68]. They obtained the relative theta power and interactions between the theta, alpha, and beta power as monitory biomarkers of motor recovery. In a currently ongoing study, EEG has been employed as a biomarker able to predict rehabilitation outcomes, providing a novel individual strategy of rehabilitation protocol (based on action observation treatment) for chronic stroke outpatients; no results have been publicized yet, but the idea appears to be promising [69]. Interesting conclusions have arisen from a study using MEG to assess cortical gamma synchronization (>30 Hz), which proved to be a predictor of recovery in patients undergoing intensive rehabilitation after stroke [70]. The next method useful as a post-stroke recovery biomarker is the assessment/presence of motor evoked potential (MEP) recorded by surface electromyography from muscles in response to TMS (transcranial magnetic stimulation). TMS is a non-invasive brain stimulation technique that allows to investigate as well as to modulate cortical excitability. There is clearly an understandable agreement that the presence of MEP (due to TMS) predicts good motor recovery as well as shorter MEP latencies, which means shorter central motor conduction times are associated with improved outcome [71]. In a retrospective study from Korea, 113 participants underwent TMS-induced MEP to estimate corticospinal excitability within 3 weeks after stroke onset [72]. They confirmed the MEP responsiveness value as one of the strong instruments of predicting motor function at 3 months after stroke. The next valuable method is to evoke cortical activity by TMS and afterwards measure cortical reorganization by EEG. Pellicciari et al. proved that TMS-evoked alpha oscillatory activity recorded just after stroke was associated with better functional recovery at 40- and 60-day follow-up evaluations [73]. Therefore, the power of the alpha rhythm is not only to predict recovery, but could also be helpful in determining the temporal window for enhancing neuroplasticity.

**Table 1 jcm-11-02473-t001:** Selected neuroimaging biomarkers of stroke recovery.

Biomarker	Type of Imaging	Usefulness Depending on the Stroke Phase	References
MRI-DTI (diffusion tensor imaging)	assess white matter integrity	acute, subacute, chronic	[42,44,45,48]
Ultra-short echo time MRI angiography	visualize macro- and microvasculature	acute, subacute	[34]
Steady-state contrast-enhanced MRI	assess vascular reorganization	subacute, chronic	[30]
Dynamic contrast-enhanced MRI	assess blood-brain barrier integrity	acute, subacute	[31]
Resting-state functional MRI	functional connectivity	subacute	[57,58]
Magnetic Resonance Spectroscopy	assess metabolic changes	subacute, chronic	[62,63]
EEG (electroencephalography)	assess balance between excitatory and inhibitory cortical actions	acute, subacute, chronic	[65,66,67]
TMS (transcranial magnetic stimulation) with MEP (motor evoked potential)	assess motor corticospinal excitability	subacute, chronic	[71,72]
TMS with EEG	assess cortical reorganization	subacute	[73]

## 4. Future Directions—Multimodal Panels of Neuroimaging Biomarkers and Application of Machine Learning Models

The aim of this article is not only to review neuroimaging stroke recovery biomarkers but also to analyze their application in treatment strategies. Therefore, it seems to be most beneficial to categorize individuals for personalized treatment based on multimodal panels including various biomarkers of stroke recovery. Recently introduced by Stinear et al., the Predict Recovery Potential (PREP) algorithm suggests combining clinical scales, transcranial magnetic stimulation, and diffusion-weighted magnetic resonance imaging to provide accurate predictions of upper limb function [71]. This information was used to modify therapy and enhance rehabilitation efficiency. Firstly, the Medical Research Council grades for shoulder abduction and finger extension strength within 3 days of stroke symptom onset were assessed. For patients with a summed score below 8 (out of 10), the functional integrity of the corticospinal tract was evaluated by identification of the presence or absence of paretic upper limb MEPs using TMS. The presence of MEPs indicated good prognosis. For individuals without MEPs, the characteristics of corticomotor pathways were assessed with diffusion tensor imaging. Low asymmetry in FA of the corticospinal tract (greater integrity of the ipsilesional corticospinal tract) was correlated with a prognosis of limited functional improvement; and high asymmetry in FA of the corticospinal tract (less integrity of the ipsilesional corticospinal tract) was correlated with the poorest prognosis for functional improvement. In this way, the algorithm predicts one of four possible outcomes for each patient: excellent, good, limited, or none. In accordance with this stratification, a suitable rehabilitation strategy was performed. Based only on clinical scales (without biomarker information about the corticospinal tract), the potential for a good recovery of function might go unrecognized by clinicians. Referring to multimodal panels of stroke recovery biomarkers, it is necessary to mention a very promising ongoing study of Picelli et al., where the study protocol for a randomized controlled trial is being developed [74]. The study will allow definition of a set of biomarkers (including neuroimaging: diffusion tensor imaging (DTI) and functional magnetic resonance imaging (fMRI); and neurophysiological: motor evoked potentials (MEP) and intracortical excitability measured by single and paired transcranial magnetic stimulation (TMS); somatosensory evoked potentials (SEP); and brain connectivity measured by electroencephalographic (EEG) phase synchrony) related to stroke recovery and rehabilitation result in order to discover patients with stronger potential for improvement and define personalized rehabilitation programs.

Future directions towards predicting post-stroke recovery and treatment outcomes could be machine-learning algorithms, which classify and predict the participants’ responsiveness to therapy [75]. In this pilot study, 64 post-stroke aphasia patients were analyzed, the predictive framework was created based on collected data—demographic, brain structure (MRI) and behavioral (clinical scales of aphasia severity)—and then Random Forest models were used to evaluate the importance of these features. Preliminary results of this study suggest the potential of their framework to predict individualized rehabilitation. Furthermore, the utility of machine learning approaches has been newly explored to predict post-stroke recovery relying on multi-channel electroencephalographic recordings and clinical scales [76]. In this study, nonlinear support vector regressor (SVR) was exploited to predict recovery in the acute phase of stroke, which might allow early and personalized therapies.

## 5. Conclusions

The use of neuroimaging techniques allows prediction of recovery potential more objectively, based not solely on functional clinical scales which represent a measure of impairment. However, it is relevant for stroke recovery to also consider other aspects, such as age, gender, type of stroke, type of treatment, as well as environmental factors. The importance of these factors should be assessed individually, but very often it is impossible to isolate one critical variable in clinical studies and therefore heterogenous groups might cancel individual benefits. To overcome this challenge, multimodal panels of biomarkers defining patients with greater recovery potential appear to be an appealing tool. Likewise, understanding the processes of angiogenesis and neuroplasticity that occur during stroke recovery, as well as monitoring these processes (by neuroimaging and neurophysiological techniques), is extremely important for planning effective treatment. Mentioned above, research using various types of biomarkers (clinical, imaging, neurophysiological, and genetic-molecular) is very promising for defining stroke rehabilitation protocol in future studies [74]. To further improve poststroke recovery, neuroimaging techniques may also be implemented to enhance robot-assisted rehabilitation [77]. This allows for monitoring of the rehabilitation process, providing precise feedback from therapy and adapting to the specific needs of patients. Nevertheless, there are still important challenges to overcome in future studies. Considerations should include issues such as suitable sample sizes, standardized methods and patient stratification. Neuroimaging and neurophysiological techniques are already well-proven as diagnostic tools for stroke but still require establishment of their prognostic value for stroke recovery.

## Figures and Tables

**Figure 1 jcm-11-02473-f001:**
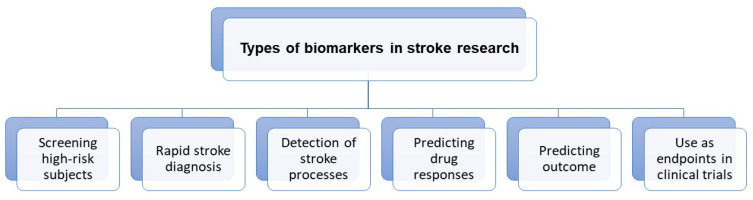
Roles of stroke biomarkers.

**Figure 2 jcm-11-02473-f002:**
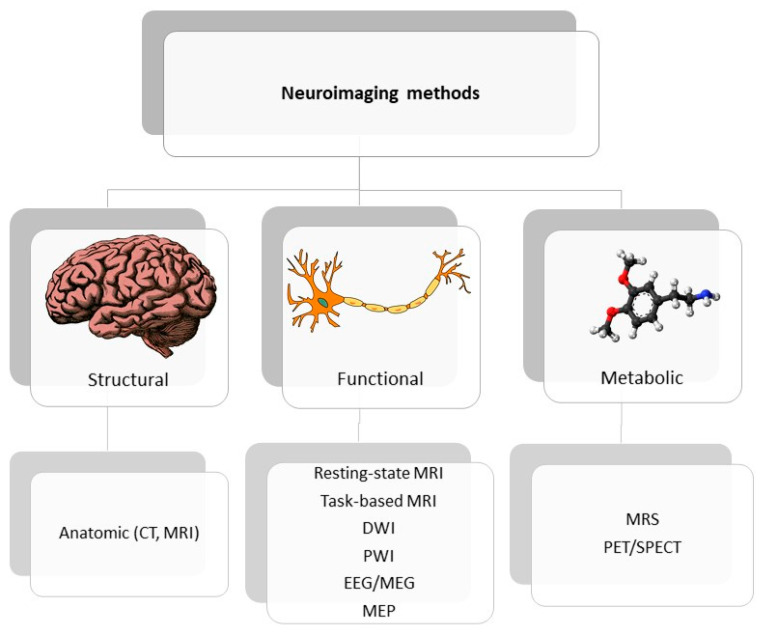
Classification of neuroimaging methods; CT—computer tomography, EEG—electroencephalography, MEP—motor evoked potential, MRI—magnetic resonance imaging, MRS—magnetic spectroscopy, PET/SPECT—positron emission tomography/ single photon emission computed tomography.

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
