# Peer review of "Neuroimaging Techniques as Potential Tools for Assessment of Angiogenesis and Neuroplasticity Processes after Stroke and Their Clinical Implications for Rehabilitation and Stroke Recovery Prognosis"

_jcm, 2022, doi:10.3390/jcm11092473_

Round 1

Reviewer 1 Report

The authors provide a very detailed overview of the use of modern imaging modalities to predict clinical outcome of patients with acute strokes. This is an overview of the state of current developments on this topic; this should be clearly stated in the text ("review of the literature") as well as in the introduction, since the authros do not present their own results or new data. Overall, the overview is good, but also very long. I recommend shortening, e.g., in the introduction (it is assumed that readers know what a stroke means clinically).  

Author Response

Dear Reviewer, 

Thank You very much for your kind reply, and also for taking the time to help us further improve our manuscript. 

In the text below we have included responses to your comments. 

The Reviewer: „ The authors provide a very detailed overview of the use of modern imaging modalities to predict clinical outcome of patients with acute strokes. This is an overview of the state of current developments on this topic; this should be clearly stated in the text ("review of the literature") as well as in the introduction, since the authros do not present their own results or new data. „ 

The authors answer: Thank You very much for your positive opinion and suggestions. The text and introduction has been changed in accordance to your advice, and now it is as below: 
Abstract: „In this review article we introduce neuroimaging techniques dedicated to stroke recovery analysis with reference to angiogenesis and neuroplasticity processes.” 
Introduction: „The aim of this article is to review the literature and analyze neuroimaging measurements and its role for predicting recovery after stroke. „ 
Third section title: „Stroke recovery prognosis based on selected neuroimaging measurements – review of literature.” 

The Reviewer: „Overall, the overview is good, but also very long. I recommend shortening, e.g., in the introduction (it is assumed that readers know what a stroke means clinically).   

The authors answer: Thank You very much for this suggestion. The part of introduction about clinical symptoms has been shorten, and now it is as below: „Stroke is an acute cerebral, spinal, or retinal vascular accident with neurological dysfunction, that persist longer than 24 h or one of any duration when infarction or hemorrhage corresponding to symptoms is demonstrated by imaging (computed tomography/ magnetic resonance scans) or autopsy [1]. The clinical symptoms are very heterogenous and conditional on the topography of damage [2].  

Sincerely yours,  
Lidia Wlodarczyk and co-Authors 

Reviewer 2 Report

WÅ‚odarczyk and colleagues in their work titled “Neuroimaging techniques as potential tools for assessment of angiogenesis and neuroplasticity processes after stroke and their clinical implications for rehabilitation and stroke recovery prognosis.” provide a review of current neuroimaging techniques and their current or potential role in predicting post-stroke recovery.

This is an important, active and growing field. The techniques and methods discussed provide the necessary information for the optimization of individual treatment in order to gain maximal patient benefit.

I have some comments:

-In the introduction (line 56 onwards) “neuromodulation” should be mentioned as one possible intervention (e.g. 10.1177/1545968315586464, https://doi.org/10.1007/s40141-020-00257-5, https://doi.org/10.1007/s00422-020-00818-w, https://doi.org/10.1016/j.neurom.2021.12.002).

-Magnetoencephalography (MEG) is only mentioned in passing later in the text. It should be included in Figure 2 and the text.

-The sentence “It’s considered that neuronal activation and cerebral blood flows are coupled” (Line 237) might be followed by a citation such as: e.g. this book: fMRI: From Nuclear Spins to Brain Functions)

Author Response

Dear Reviewer, 

Thank You very much for your kind reply, and also for taking the time to help us further improve our manuscript. 

In the text below we have included responses to your comments. 

The Reviewer: „In the introduction (line 56 onwards) “neuromodulation” should be mentioned as one possible intervention (e.g. 10.1177/1545968315586464 , https://doi.org/10.1007/s40141-020-00257-5, https://doi.org/10.1007/s00422-020-00818-w, https://doi.org/10.1016/j.neurom.2021.12.002)." 

The authors answer: Thank You very much for this essential suggestion. „Neuromodulation” has been added as a possible intervention and now it is as below:  When considering the strategies of modern post-stroke rehabilitation, the importance of non-invasive neuromodulation techniques including repetitive transcranial magnetic stimulation and transcranial direct current stimulation should be also emphasized [9-12]. 

The Reviewer: „Magnetoencephalography (MEG) is only mentioned in passing later in the text. It should be included in Figure 2 and the text.” 

The authors answer: Thank you very much for this valuable remark. We have included magnetoencephalography (MEG) in Figure 2, we have also expanded the part of the article about MEG, in addition to the Zappasodi study [66] we have also presented Pellegrino results [71] and now it is as below: „ The general findings, suggesting bad recovery in post-stroke patients investigated with EEG or MEG at the acute or subacute stages, indicate predominant inhibitory processes in the peri-lesional areas of cortex, that was shown by increased low-frequency oscilla-tions [66,67]. Simultaneously, good recovery after stroke is associated with increased sensorimotor excitability in acute stages, that was presented with lower beta-rebound as a reaction to tactile finger stimulation [68]. Regarding the chronic post-stroke stag-es, in Mane’s study, changes in the cortical activity after different types of motor reha-bilitation and its possible outcome using the Quantitative Electroencephalography (QEEG) features were analyzed [69]. They obtained the relative theta power and inter-actions between the theta, alpha, and beta power as monitory biomarkers of motor re-covery. In currently ongoing study, EEG has been employed as a biomarker able to predict rehabilitation outcomes and providing a novel individual strategy of rehabili-tation protocol (based on Action Observation Treatment) for chronic stroke outpa-tients, no results are publicized yet, but idea appears promising [70]. Interesting con-clusions arise from a study with the use of MEG to assess cortical gamma synchroniza-tion (> 30 Hz), which proved to be a predictor of recovery in patients undergoing in-tensive rehabilitation after stroke [71].” 

The Reviewer: „The sentence “It’s considered that neuronal activation and cerebral blood flows are coupled” (Line 237) might be followed by a citation such as: e.g. this book: fMRI: From Nuclear Spins to Brain Functions) 

The authors answer: Thank You very much, the citation has been added. 

Sincerely yours,  
Lidia Wlodarczyk and co-Authors 

Round 2

Reviewer 1 Report

Thanks for making changes accordining to my suggestions.